# Correlation between 25-hydroxyvitamin D/D3 Deficiency and COVID-19 Disease Severity in Adults from Northern Colorado

**DOI:** 10.3390/nu14245204

**Published:** 2022-12-07

**Authors:** Bridget A. Baxter, Michaela G. Ryan, Stephanie M. LaVergne, Sophia Stromberg, Kailey Berry, Madison Tipton, Nicole Natter, Nikiah Nudell, Kim McFann, Julie Dunn, Tracy L. Webb, Michael Armstrong, Nichole Reisdorph, Elizabeth P. Ryan

**Affiliations:** 1Department of Environmental Radiological Health Science, College of Veterinary Medicine and Biomedical Sciences, Colorado State University, Fort Collins, CO 80523, USA; 2Medical Center of the Rockies, University of Colorado Health, Loveland, CO 80538, USA; 3Department of Clinical Sciences, Colorado State University, Fort Collins, CO 80523, USA; 4Skaggs School of Pharmacy and Pharmaceutical Sciences, University of Colorado Anschutz Medical Campus, Aurora, CO 80045, USA

**Keywords:** COVID-19, Vitamin D, 25-hydroxyvitamin D, ergocalciferol D2, cholecalciferol D3, deficiency

## Abstract

Vitamin D deficiency is common in the United States and leads to altered immune function, including T cell and macrophage activity that may impact responses to SARS-CoV-2 infection. This study investigated 131 adults with a history of a positive SARS-CoV-2 nasopharyngeal PCR and 18 adults with no COVID-19 diagnosis that were recruited from the community or hospital into the Northern Colorado Coronavirus Biorepository (NoCo-COBIO). Participants consented to enrollment for a period of 6 months and provided biospecimens at multiple visits for longitudinal analysis. Plasma 25-hydroxyvitamin D levels were quantified by LC-MS/MS at the initial visit (n = 149) and after 4 months (n = 89). Adults were classified as deficient (<30 nM or <12 ng/mL), insufficient (<30–50 nM or 12–20 ng/mL), or optimal (50–75 nM or >20 ng/mL) for 25-hydroxyvitamin D status. Fisher’s exact test demonstrated an association between disease severity, gender, and body mass index (BMI) at baseline. Mixed model analyses with Tukey-Kramer were used for longitudinal analysis according to BMI. Sixty-nine percent (n = 103) of the entire cohort had optimal levels of total 25(OH)D, 22% (n = 32) had insufficient levels, and 9% (n = 14) had deficent levels. Participants with severe disease (n = 37) had significantly lower 25-hydroxyvitamin D (total 25(OH)D) when compared to adults with mild disease (*p* = 0.006) or no COVID-19 diagnosis (*p* = 0.007). There was 44% of the cohort with post-acute sequalae of COVID-19 (PASC) as defined by experiencing at least one of the following symptoms after 60 days’ post-infection: fatigue, dyspnea, joint pain, chest pain, forgetfulness or absent-mindedness, confusion, or difficulty breathing. While significant differences were detected in 25-hydroxyvitamin D status by sex and BMI, there were no correlations between 25-hydroxyvitamin D for those without and without PASC. This longitudinal study of COVID-19 survivors demonstrates an important association between sex, BMI, and disease severity for 25-hydroxyvitamin D deficiency during acute stages of infection, yet it is not clear whether supplementation efforts would influence long term outcomes such as developing PASC.

## 1. Introduction

Vitamin D is a fat-soluble compound naturally present in foods, used to fortify foods, and available as a dietary supplement in two main forms: D2 and D3 (also known as Ergocalciferol and Cholecalciferol, respectively). It is also produced endogenously from sunlight by the skin. Vitamin D absorbed by the intestines and synthesized by the skin as well as serum concentrations of 25-hydroxyvitamin D are indicators of controbutions across sources [1,2]. Vitamin D metabolite receptors reside in many cells, and as a whole, the 1, 25-hydroxyvitamin D network impacts a suite of host processes and cellular functions [1]. Vitamin D metabolism influences immune cells in a manner that decreases risk for developing chronic diseases such as diabetes and cardiovascular disease [3,4,5,6,7]. The effects of Vitamin D on regulation of the NF-kB pathway in immune cells may also be important regulators during infection and for the production of antimicrobial peptides and cytokines [8,9]. Reduced levels of Vitamin D have been associated with increased susceptibility and severity of bacterial and viral infections [8]. Vitamin D also assists in endotoxin tolerance and appears to reduce secondary bacterial infections [5]. 

Nearly one-fourth of the United States population is deficient in Vitamin D and at risk of being deficient [6]. Studies have shown this number may be even higher in the elderly and home care residents [7]. In addition to age, metabolism and bioavailability differ as a function of body mass index (BMI) and sex [10,11]. Similarly, obesity and sex differences have been identified as risk factors for COVID-19 disease severity and development of post-acute sequelae of COVID-19 infection (PASC) [12,13]. These findings, as well as the prevalence of Vitamin D deficiency, suggest that Vitamin D may serve as an underlying risk factor for COVID-19 disease severity, development of PASC, and a potential treatment and/or preventive [14,15]. 

The Northern Colorado Coronavirus Biobank (NoCo-COBIO) is a resource of clinically integrated and quality-controlled longitudinal samples collected to assess molecular markers of disease severity and relationships to PASC. The primary objective of this study was to evaluate body weight- and sex-related differences in plasma 25-hydroxyvitamin D and D3 levels and their relationship to COVID-19 disease severity and development of PASC. 

## 2. Materials and Methods

### 2.1. Study Design

One hundred and thirty-one participants diagnosed with COVID-19 and 18 adults with no COVID-19 diagnosis were enrolled into the Northern Colorado Coronavirus Biobank (NoCo-COBIO) that was established at Colorado State University (CSU) as part of a community-based collaboration with the University of Colorado Health (UCHealth) in Fort Collins, Colorado. Inclusion criteria for all participants included (1) a positive SARS-CoV-2 polymerase chain reaction (PCR) test and (2) at least 18 years of age. Patients were excluded if they were pregnant, incarcerated, or had a COVID-19 diagnosis by antigen or antibody testing. The recruitment of participants took place through UCHealth Northern Colorado hospitals, including Poudre Valley Hospital (PVH) in Fort Collins, Medical Center of the Rockies (MCR) in Loveland, and Greeley Hospital in Greeley. Outpatient/non-hospitalized patients were recruited via emails, recruitment flyers, and health department screening. This biorepository and written informed consent were approved by CSU’s Institutional Review Board (IRB; protocol ID 2105 (and 20-10063H) and the UCHealth IRB (Colorado Multiple IRB 20–6043) and is registered at ClinicalTrials.gov (NCT04603677). Written informed consent was obtained from all participants prior to enrollment. This biorepository is maintained in accordance with the 1964 Helsinki Declaration and its 2013 amendments. The 131 diagnosed with COVID-19 and 18 individuals with no COVID-19 diagnosis completed their study visits between July 2020 and March 2021. Cash compensation of $25 was given to all participants at each study visit. Figure 1 shows the CONSORT flow diagram for this study.

### 2.2. Participants

Participant demographics were obtained at the study visit or from electronic hospital records, and all data is stored in the password protected Research Electronic Data Capture (REDCap) web application. Participant disease severity was categorized based on the Yale Impact Score: mild (no oxygen required), moderate (1-5L oxygen requirement), and severe (greater than 5L oxygen requirement) [16]. Body mass index (BMI) was calculated using self-reported height and weight for non-hospitalized (n = 72) participants at clinic visits and electronic health records from hospitalized (n = 77) participants [15,17]. Body mass index was broken down into four categories: (1) underweight (<20), (2) normal weight (<20–24.9), (3) overweight (25–29.9), and (4) obese (>30). A symptom survey was administered at every study visit to identify PASC. In accordance to the World Health Organization (WHO) guidelines, participants were defined as having PASC if experiencing at least one of the following symptoms for at least 60 days post-infection: fatigue, dyspnea, joint pain, chest pain, forgetful or absent-mindedness, confusion, or difficulty breathing [15,18]. In addition to the COVID-19 survivors, adults without a COVID-19 diagnosis were enrolled and underwent an identical study sequence for biospecimen collection. In this present study, time of enrollment is the initial visit, and month 4 is an average of 110 days from time of enrollment; participants are defined as having PASC obtained from the completed symptom survey at month 4. All data was de-identified, and each participant was assigned a unique identifier. 

### 2.3. Biospecimen Collection

Participants were enrolled for a period of 6 months and consented to provide biospecimens at 4 visits: baseline, approximately one month after baseline (visit 2), 4 months after baseline (visit 3), and 6 months after baseline (visit 4). Participants also had the option to consent to a one-year follow up (visit 5). Study visits were conducted at the Human Performance Clinical Research Laboratory on CSU campus or in the hospital. Participants provided approximately 45 mL of blood collected into five 8 mL sodium citrate cell preparation tubes (CPT) (BD BioSciences, Franklin Lakes, NJ, USA) and one 5 mL serum separator tube (VWR). Each participant sample was de-identified and transported to the Ryan Lab on CSU campus for processing and final storage, as previously described [15]. 

The CPTs were centrifuged at 1500× *g* at room temperature for 30 min with the brake off. The plasma was removed and aliquoted out into 10 one mL tubes and stored at −80 °C until analysis. 

### 2.4. Materials

25-Hydroxyvitamin D2, 25-Hydroxyvitamin D3 and d6-25-Hydroxyvitamin D3 (deuterium-labeled internal standard), were obtained from Cerilliant (Round Rock, TX, USA). Formic acid, liquid chromatography mass spectrometry (LC/MS) grade methanol, ethanol and acetonitrile were obtained from Fisher Scientific (Fairlawn, NJ, USA). High-performance liquid chromatography (HPLC) grade water was obtained from Burdick and Jackson (Morristown, NJ, USA).

### 2.5. Preparation of Calibration Standards

Combined stock standards of 25-Hydroxyvitamin D2 and 25-Hydroxyvitamin D3 were prepared at 25 μg/mL in ethanol and stored as 100μL aliquots at −70 °C until use. The d6-25-Hydroxyvitamin D3 internal standard solution was prepared at 2.5 μg/mL and stored as 1 mL aliquots at −70 °C until use.

Immediately before use, the combined stock standard was removed from the freezer, allowed to reach room temperature, and diluted 1:4 in ethanol. An additional 7 calibration spike standards were prepared by 1:1 serial dilution with ethanol from 5 μg/mL down to 39 ng/mL. The calibration standards were prepared by adding 10 μL of calibration spike standard, 10 μL of internal standard, 100 μL of PBS, and 400 μL of 1% formic acid in acetonitrile and vortexed for 10 s.

### 2.6. Preparation of Standards and Samples

50 μL of plasma was added to a 2.0 mL microfuge tube and diluted 1:1 with PBS. 10 μL of internal standard and 400 μL of 1% formic acid in acetonitrile was added and the sample was vortexed for 10 s. The sample was then centrifuged at 14 K RPM for 10 min at 4 °C. The entire sample was then loaded onto a Captiva enhanced matrix removal (EMR) cartridge (Agilent Technologies, Santa Clara, CA, USA) and placed on a 48-position positive pressure solid phase extraction (SPE) manifold (Biotage, Charlotte, NC, USA). Vacuum was applied at 1–2 psi, and the samples were eluted into a 1.8 mL amber screw cap autosampler vial. 

### 2.7. LC-MS

Separation of 25-Hydroxyvitamin D2, 25-Hydroxyvitamin D3, and d6-25-Hydroxyvitamin D3 was performed on an Agilent 1260 series HPLC with an Agilent Poroshell EC C-18 3.0 × 50 mm 2.7 μm. Buffer A consisted of 0.1% formic acid in water, and buffer B consisted of 0.1% formic acid in methanol. Twenty microliters of the extracted sample were analyzed using the following gradient at a flow rate of 0.5 mL/min: hold at 75% B for 0.5 min, then 75% B to 98% B from 0.5 to 4 min, hold at 98% B from 4 to 6 min. The column was then re-equilibrated at 75% B for 2 min. The column temperature was held at 30 °C for the entire gradient. Mass spectrometric analysis was performed on an Agilent 6490 triple quadrupole mass spectrometer with an electrospray source in positive ionization mode. The drying gas was 250 °C at a flow rate of 11 L/min. The nebulizer pressure was 45 psi. The sheath gas temperature was 325 °C at a flow rate of 11 L/min. The capillary voltage was 5000 V. Data for 25-Hydroxyvitamin D2, 25-Hydroxyvitamin D3 and d6-25-Hydroxyvitamin D3 was acquired in multiple reaction monitoring (MRM) mode using experimentally optimized conditions obtained by flow injection analysis of authentic standards. 

### 2.8. Data Analysis

Calibration curves for 25-Hydroxyvitamin D2 and 25-Hydroxyvitamin D3 with d6-25-Hydroxyvitamin D3 as the deuterium-labeled internal standard were constructed using Agilent Masshunter Quantitative Analysis software. Results for plasma samples were quantitated using these calibration curves to obtain the concentration in ng/mL.

### 2.9. Statistical Analysis

Data were tested for the distributional assumption of normality. Total 25(OH)D (ng/mL) and 25(OH)D3 (ng/mL) without a transformation proved closest to a normal distribution. ANOVA with a Tukey-Kramer *p*-value adjustment was used to test the relationship between disease severity and total 25(OH)D (D2 and D3 together) as well as 25(OH)D3 at baseline and month 4. Data were presented as Mean ± SD. Total 25(OH)D and 25(OH)D3 were adjusted into three categories: (1) 30 nM or <12 ng/mL deficient, (2) 30–50 nM or 12–20 ng/mL insufficient, and (3) 50–75 nM or >20 ng/mL optimal [7,19,20,21,22]. There was one participant with total 25(OH)D and 25(OH)D3 above 70 ng/mL that was added to optimal category. Fisher’s exact was used to analyze associations between total 25(OH)D and D3 categories and COVID-19 disease severity. Data were presented as frequency (%). Mixed model longitudinal data analyses were used to test for main effects of BMI categories and differences in slopes for total 25(OH)D and 25(OH)D3. Mixed model longitudinal data analyses were also used to test for main effects of disease severity categories and differences in slopes for total 25(OH)D and 25(OH)D3 with PASC. Adjusted *p*-values were used for comparisons between visits and the categories. Adjusted *p*-values < 0.05 were considered significant. All analyses were performed using SAS 9.4 (Cary, NC, USA).

## 3. Results

### 3.1. Participant and Longitudinal Biorepository Cohort Demographics

Plasma total 25(OH)D and 25(OH)D3 were available for 131 adult participants diagnosed with COVID-19 and 18 adults with no COVID-19 diagnosis. Ninety adults diagnosed with COVID-19 were enrolled into the biobank in the acute phase (days 1–15 post SARS-CoV-2 PCR+) with 20 experiencing mild disease, 35 with moderate disease, and 35 with severe disease. The mean age of 57 ± 17.1 and mean BMI 31.6 ± 9.9 was established with sex distribution of 51% female and 49% male,; and 80% were hospitalized (n = 72). Forty-one adults diagnosed with COVID-19 were enrolled in the convalescent phase (days 21–208 post SARS-CoV-2 PCR+). From the convalescent participants, there were 35 with mild disease, 5 with moderate disease, and 1 diagnosed with severe disease. This group had a mean age of 46.1 ± 17.9, mean BMI 28.1 ± 5.9, and 67% female, and 17% hospitalized (n = 7). Eighteen adults with no COVID-19 diagnosis with mean age of 47.8 ± 10.1 and mean BMI 25.0 ± 5.3 were 88% female. Eight participants diagnosed with COVID-19 and two participants without COVID-19 diagnosis reported taking Vitamin D supplementation (Table 1). Plasma total 25(OH)D and D3 were analyzed for 149 participants [18 uninfected (no COVID-19 diagnosis), 55 mild, 39 moderate, 37 severe diagnosis]. Sixty-nine percent (n = 103) of the entire cohort had optimal levels of total 25(OH)D, 22% (n = 32) insufficient, and 9% (n = 14) had deficient levels. Figure 2 shows plasma total 25(OH)D for adults with and without COVID-19 and indicates presence or absence of pre-existing conditions. Participants with severe disease had lower total 25(OH)D and 25(OH)D3 status than those who had no COVID-19 diagnosis (*p* = 0.007). There are more individuals with pre-existing conditions in the severe (78%) and moderate (75%) disease groups when compared to the mild (22%) disease group. Appendix A illustrates mild, moderate, and severe disease with and without pre-existing conditions.

Fourteen percent of adults diagnosed with severe infection were deficient (<12 ng/mL), 44% were insufficient (12–20 ng/mL), and 42% had optimal (>20 ng/mL) levels of total 25(OH)D and 25(OH)D. Twelve percent % of adults diagnosed with moderate infection were deficient, 25% were insufficient, and 63% had optimal levels of total 25(OH)D and D3. Seven percent of adults diagnosed with mild infection were deficient, 18% insufficient, and 75% had optimal levels of 25(OH)D and D3. Seventy-eight percent of adults without COVID-19 were insufficient, and 22% had optimal levels of total 25(OH)D and D3 (Appendix A). All adults without a history of COVID-19 had optimal levels of total 25 (OH)D.

### 3.2. 25-hydroxyvitamin D and D3 Differences by Sex at Time of Enrollment

Table 2 shows baseline plasma 25(OH)D and D3 levels for 131 COVID-19 infected adult males and females (55 mild, 40 moderate, and 36 severe). In this cohort, females (n = 72) had higher levels of total 25(OH)D status when compared to males (n = 59) at time of enrollment (*p* = 0.037).

### 3.3. 25-hydroxyvitamin D and D3 by Disease Severity and PASC

There were no significant changes in status from time of enrollment to month 4 in total 25(OH)D and D3 for any disease severity category nor for participants with no COVID-19 diagnosis (Appendix A). No correlations were detected between total 25(OH)D and PASC. However, severity of disease categories did have a small effect on total 25(OH)D status (*p* = 0.044). There was a difference in the main effect of disease severity for adults diagnosed with severe and moderate COVID-19 disease with lower total 25(OH)D plasma samples at baseline and 4 months (*p* = 0.0179). There was no effect of time points (*p* = 0.603), and no differences in slopes of total 25(OH)D over time for any of the disease severity categories (*p* = 0.2628) (Figure 2).

Figure 3A shows total plasma 25(OH)D values for 84 participants that completed two study visits: at time of enrollment and month 4 by disease severity and PASC status. Figure 3B shows plasma total 25(OH)D values for the 44 participants who were enrolled in the acute phase (day 1–14 post PCR+) and at day 83–140 post PCR+. The left panel shows participants trending toward increased total 25(OH)D, and the right panel shows participants trending to decrease between days 1–14 and then 83–140. Forty-eight percent of the 44 participants had decreased total 25(OH)D concentrations after COVID-19, although there was no statistically significant correlation between total 25(OH)D and developing PASC.

### 3.4. 25-hydroxyvitamin D and D3 Differences by Body Mass Index

Table 3 highlights plasma total 25(OH)D and D3 levels at baseline from COVID-19 infected adults separated according to BMI (normal/underweight, overweight, and obese). Lower total 25(OH)D and 25(OH)D3 levels were associated with obesity when compared to normal/underweight weight. Plasma total 25(OH)D and D3 from 84 adults with a COVID-19 diagnosis at time of enrollment and at month 4 was assessed with the use of mixed model analyses and Tukey-Kramer *p*-value correction (Appendix A). There was a main effect of BMI category for total 25(OH)D and D3 (*p* = 0.0009). After adjusting for multiple comparisons, total 25(OH)D and D3 was lower in obese participants compared to normal/underweight participants for baseline only (*p* = 0.048). There was no significant change from time of enrollment and month 4 in total 25(OH)D or D3. Appendix A shows all total 25(OH)D, D2, and D3 concentrations for the entire cohort. However, Appendix A shows five participants (3%) who had 25(OH)D2 concentrations detected above the limit of quantification (LOQ) that contributed to the total 25(OH)D.

## 4. Discussion

Numerous studies have confirmed the importance of sufficient levels of Vitamin D for overall health and disease prevention [23,24,25]. Vitamin D is metabolized and used by virtually every cell in the body via the essential Vitamin D Receptor (VDR), resulting in alterations in immune regulation and inflammation through epigenetics and gene expression [26,27]. Findings from this observational cohort investigation of adults with and without COVID-19 (n = 149) showed that 36% (n = 54) of adult Northern Coloradoans were deficient or insufficient (<20 ng/mL) in Vitamin D between July 2020 and July 2021, [21]. It remains unknown if illness alters total 25-hydroxyvitamin D levels, but in several studies on critical illness, deficiency led to worse outcomes [28]. In this study cohort, participants with severe disease had significantly lower total 25-hydroxyvitamin D status than those with and mild or no disease; no statistical difference was observed between severe and moderate disease, nor was there a relationship for developing PASC.

Deficiencies in total 25(OH)D have been correlated with risk for other infections including influenza and tuberculosis [29]. Emerging evidence exists in the UK and Portugal for potential causal relationships between Vitamin D status and COVID-19 disease risk [3,5,30]. Evidence supports an association between Vitamin D deficiency and incidence of pulmonary exacerbations in chronic airway diseases such as asthma, yet results from supplementation studies evaluating prevention or amelioration of these exacerbations are inconsistent [8]. Interestingly, data indicates that Vitamin D-dependent processes are not linearly related to plasma 25(OH)D levels but may show greatest impact at the most deficient Vitamin D levels. Low 25(OH)D levels are unable to provide enough substrate for effective intracrine conversion to the active form, 1, 25(OH)D2 [31]. Further exploration of the effects of Vitamin D levels on disease is warranted.

Several co-morbidities that are risk factors for COVID-19 are also associated with Vitamin D status. Cross-sectional studies conducted in the United States suggest Vitamin D deficiency is associated with an increased risk of cardiovascular disease [32], including the development of hypertension and sudden cardiac death [33,34]. Molecular mechanisms include down regulation of the renin-angiotensin-aldosterone system and direct effects on the heart [35]. Chronic obstructive pulmonary disease (COPD) is another pre-existing health condition in adults associated with low Vitamin D levels. Both cardiovascular disease and COPD were increased in prevalence when compared to adults without history of SARS-CoV-2 infection [32,36]. Individuals with COPD are at high risk for experiencing severe symptoms and PASC from COVID-19 after infection, and it has been shown that supplementation of Vitamin D in patients with COPD reduced the exacerbations of symptoms, possibly triggered by infections [36]. Likewise, Vitamin D has been found to prevent hypermethylation in diabetes mellitus, an epigenetic alteration shown to be prevalent in multiple gene promoter regions of many diabetes-related genes [37]. Vitamin D plays a significant role in maintenance of the epigenome. For example, the effects of epigenetic modifications of DNA on regulation of ACE2 expression were established before the emergence of SARS-CoV-2 [38]. In preventing hypermethylation, Vitamin D may prevent viral-mediated expression of the DNA demethylases, thus preventing viral entry [39]. Taken together, Vitamin D may be a link between these co-morbidities and worse COVID-19 outcomes, yet requires further study at the intracrine and autocrine level.

Notably, no correlation was identified in this cohort between total 25-hydroxyvitamin D levels and risk for PASC, although other studies have made this association [40]. The variability in timing between first and second blood draws may account for this difference because it spans both acute and convalescent phases of infection. Interestingly, the uptake and release of 25(OH)D in skeletal muscle may also contribute to the stability in circulation when measured over several months [11]. Our cohort analysis revealed differences in total 25-hydroxyvitamin D status according to sex and BMI, which is consistent with prior studies [41]. Interestingly, this study measured sex differences whereby males have greater total 25-hydroxyvitamin D deficiency than females, which may impact risk for other chronic conditions [42]. Recent data suggests that muscle may be an extravascular reservoir of Vitamin D and may contribute to lower levels in men through sequestration [11]. Participants with a BMI of 25 or greater had lower total 25-hydroxyvitamin D levels, a relationship that was independent of COVID-19 disease severity. Although Vitamin D is found in adipose tissue, it can only be released when stored fatty acids are mobilized to supply energy; thus adipose tissue sequesters Vitamin D rather than providing a functional storage site [43,44,45]. Higher rates of low Vitamin D deficiency levels in males compared to females and sequestration in patients who are obese may indicate an important target population for increased Vitamin D supplementation to reduce COVID-19 disease severity.

Total 25(OH)D is a secosteroid with hormone properties that exhibits activity in all cells through VDR-dependent mechanisms and has an array of impact on genes involved in cell proliferation, differentiation, bactericidal protein production, and innate and adaptive immunity. This and other studies suggest that individuals with chronic conditions, such as obesity, hypertension, COPD, and diabetes combined with Vitamin D deficiency and insufficiency should be classified as ‘high risk’ for infection and poorer outcomes. Vitamin D supplementation may assist in reducing risk of severe disease from COVID-19 particularly for individuals with pre-existing conditions. Given the low-risk profile of Vitamin D supplementation, it may be an area of targeted intervention opportunity in patients found to have deficient or insufficient levels. Future research is needed to examine differences in total 25(OH)D between males and females for the development of PASC types, such as fatigue, respiratory and neuro-cognitive symptoms [13,17].

## 5. Limitations

Study limitations herein include small sample size and study duration. A subset of the cohort (n = 37) received convalescent plasma treatment, and 6.7% (n = 10/149) reported Vitamin D supplementation. The variation in timing of visits according to days’ post PCR+ was impacted by time of enrollment. Three hospitalized participants reported a diagnosis of diabetes after the COVID-19 diagnosis that were not included in the pre-existing conditions. Notably, these analyses for total 25(OH)D levels used blood samples collected prior to widespread vaccination for COVID-19. There are limitations to our cohort for understanding the greater magnitude of deficiency in males as relative to lifestyle, supplementation, and overall health. Notably, a stay-at-home order was implemented during the study period due to the Cameron Peak wildfire near Fort Collins, CO, USA (August–November 2020), which may have further limited outdoor activity in northern Colorado and opportunity to receive sunlight sources of Vitamin D. Changes in plasma total 25(OH)D levels can take several weeks to months to occur, and this study timeframe of 4 months between samples may be different from rechecks that are clinically recommended in 6–12 weeks.

## 6. Conclusions

We showed that plasma total 25-hydroxyvitamin D levels were lower in obese/overweight adult participants in comparison to normal weight participants and were not related to risk for developing PASC. Our findings support that lower Vitamin D levels are associated with severe disease, and we therefore put forth that supplementation of Vitamin D, with physician oversight, may reduce risk of severe COVID-19 disease for male individuals with underlying chronic diseases. In combination with vaccination, dietary supplementation of Vitamin D in obese adults remains a promising area of exploration for reducing risk and disease severity following COVID-19 infection.

## Figures and Tables

**Figure 1 nutrients-14-05204-f001:**
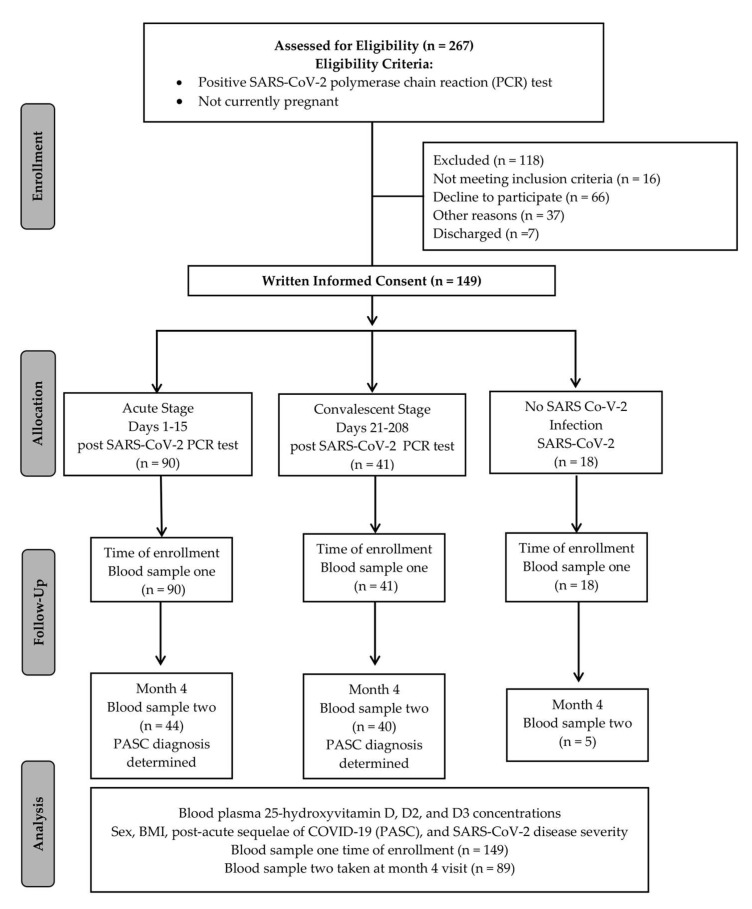
Study design for plasma total 25(OH)D and D3 analysis from 131 adults diagnosed with COVID-19 and 18 adults with no COVID-19 diagnosis. The plasma total 25(OH)D and D3 concentrations were analyzed for statistical comparison between adults with and without COVID-19 and for comparison from time of enrollment and after 4 months.

**Figure 2 nutrients-14-05204-f002:**
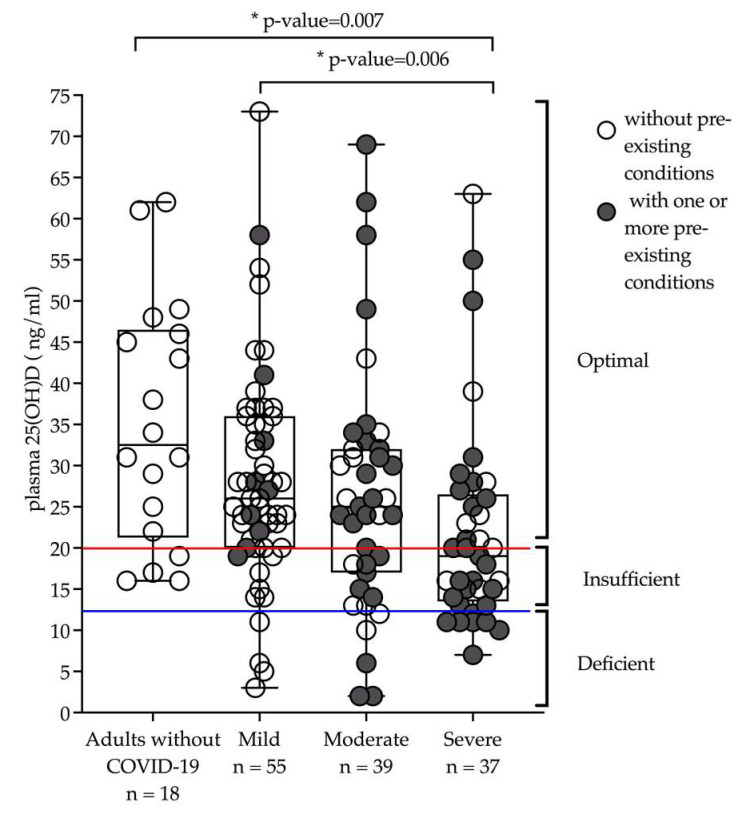
Plasma total 25(OH)D status was significantly different between participants with severe disease and those adults without COVID-19. Above the red horizontal line indicates optimal level, between the blue and red horizontal lines indicates sufficient, and below the blue horizontal lines indicates deficient. White circles indicate no co-morbidities/pre-existing chronic disease conditions, and grey circles indicate one or more co-morbidities/pre-existing chronic disease condition(s). Results were analyzed using SAS 9.4 (Cary, NC, USA). The * indicates significant total 25(OH)D *p*-value < 0.05.

**Figure 3 nutrients-14-05204-f003:**
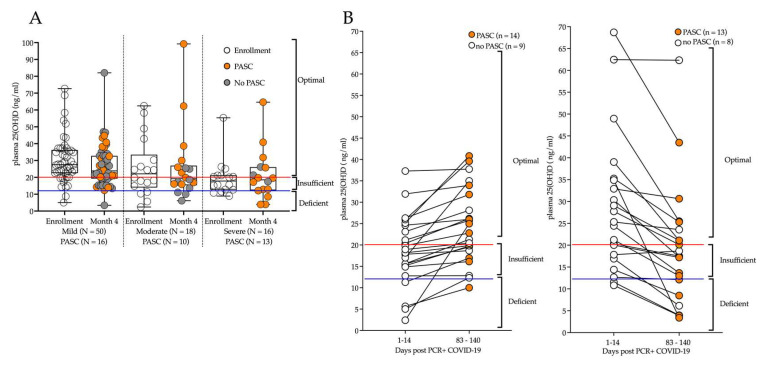
Longitudinal plasma levels of total 25(OH)D for adults diagnosed with COVID-19 at time of enrollment and after 4 months. (**A**) Plasma total 25(OH)D was compared across mild, moderate, and severe COVID-19 disease at time of enrollment; orange circles indicate PASC at month 4. (**B**) Plasma total 25(OH)D for adults enrolled in the acute phase (days 1–14 post PCR+) and at month 4 (day 83–140 post PCR+); Left panel represents participants trending to have an increase in total 25(OH)D from day 1–14 to day 83–140; Right panel shows trending reductions in total 25(OH)D from day 1–14 to day 83–140. For days 83–140, the white circles indicate no PASC, orange circles indicate PASC; below blue line indicates deficient, between blue and red line indicates insufficient, and above red line indicates optimal total 25(OH)D. No statistical significance was detected between the PASC groups or timepoints.

**Table 1 nutrients-14-05204-t001:** Participant demographics for 25-hydroxyvitamin D (n = 149).

	Days 1–15 Post PCR+	Days 21–208 Post PCR+	Adults without COVID-19
	n = 90	n = 41	n = 18
Age, years ± SD	56.5 ± 17.4	47.0 ± 18.0	47.8 ± 10.1
BMI, mean ± SD	31.8 ± 9.9	28.0 ± 6.0	25.0 ± 5.3
Sex, no. (%)			
Female	45 (50)	27 (66)	14 (88)
Male	45 (50)	14 (34)	4 (22)
Ethnicity, (%)			
Hispanic/Latinx	19 (21)	6 (15)	1 (6)
Non-Hispanic/Latinx	71 (79)	35 (85)	17 (94)
Hospitalized	71 (79)	7 (17)	-
Non-hospitalized	19 (21)	35 (83)	-
Received Convalescent Plasma	33 (37)	2 (5)	-
Vitamin D supplementation	6 (7)	2 (5)	2 (11)
Post-acute sequelae of COVID-19 (PASC)	26	12	-
Disease severity, no. (%)			
Mild	21 (23)	34 (83)	-
Moderate	35 (39)	5 (12)	-
Severe	34 (38)	2 (5)	-
Without pre-existing conditions, (%)	35 (39)	27 (66)	
With pre-existing conditions, (%)	55 (61)	14 (34)	
DM	29 (32)	3 (7)	-
COPD	8 (9)	1 (2)	-
HTN	31 (34)	6 (15)	-
Asthma	16 (18)	5 (12)	-

Note: Disease severity was determined by oxygen use in accordance with the Yale Impact Score: mild (no oxygen required), moderate (1–5 L oxygen use), severe (>5 L oxygen use). BMI and pre-existing conditions were obtained from hospital electronic records or self-reported at clinic visits. BMI = Body Mass Index, DM = diabetes mellitus, COPD = chronic obstructive pulmonary disease, HTN = hypertension, SD = standard deviation.

**Table 2 nutrients-14-05204-t002:** Total 25-Hydroxyvitamin D and 25-Hydroxyvitamin D3 differentiated by sex in adults with a history of COVID-19 (N = 131).

Vitamin	Males (N = 59)	Females (N = 72)	*p*-Value
25(OH)D	23.2 ± 12.6	28.2 ± 13.9	**0.037**
25(OH)D3	22.8 ± 12.8	27.4 ± 14.4	0.057

Values are presented in ng/mL, as estimate ± standard error (SE), *p*-values < 0.05 were considered significant. Analyses were performed using SAS 9.4 (Cary, NC, USA). Bolded *p*-value indicates significance.

**Table 3 nutrients-14-05204-t003:** Baseline total 25-Hydroxyvitamin D and 25-Hydroxyvitamin D3 by body mass index in adults with history of COVID-19 infection (N = 131).

Vitamin	Normal/Underweight (N = 30)	Overweight (N = 37)	Obese (N = 64)	Omnibus ANOVA *p*-Value
25(OH)D	29.5 ± 13.9	29.4 ± 15.2	21.4 ± 11.6 *^,a^	0.016
25(OH)D3	29.5 ± 13.9	29.4 ± 15.2	22.5 ± 11.6 *	0.004

Values are presented in ng/mL, as estimate ± standard error (SE), *p*-values < 0.05 were considered significant. Analyses were performed using SAS 9.4 (Cary, NC, USA). Bolded *p*-valued indicates significant. * indicates significant when compared to normal/underweight; ^a^ indicates significant when compared to normal/underweight and overweight.

## Data Availability

All data for this study are presented in Appendix A in the manuscript. Additional information regarding the status of the biorepository is available on the trial registration site. ClinicalTrial.gov (NCT04603677).

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
