# Peer review of "Correlation between 25-hydroxyvitamin D/D3 Deficiency and COVID-19 Disease Severity in Adults from Northern Colorado"

_nutrients, 2022, doi:10.3390/nu14245204_

Round 1
Reviewer 1 Report
This is a welcome investigation of the hypothesis that low vitamin D status is a risk factor for increased severity of COVID-19 infections. However, the findings were inconclusive as low vitamin D status was frequently found amongst the two study groups of COVID-19 infected and non-infected individuals. There are major problems throughout the text in the presentation of data concerning vitamin D status of the subjects. These are itemised as follows:
1. Line 24: The term “plasma vitamin D levels were quantified”. This is quite wrong as what was measured was 25-hydroxyvitamin D in blood plasma. It is confusing to specify “vitamin D levels” because there is vitamin D in blood as well as 25-hydroxyvitamin D. Therefore, for every measurement provided in this document the term “25-hydroxyvitamin D” should be used and not “vitamin D” because vitamin D concentration was not measured.
2. Lines 25-26: The concentration of 25-hydroxyvitamin D in blood is grouped in the so-called ranges of “<30 ng/ml” “50 ng/ml” and “51-71 ng/ml”. The units here are incorrect. These values are not ng/ml but instead are “nmoles/litre” or “nM” The values expressed in ng/ml to indicate deficiency or sufficiency of vitamin D status are not those quoted. If the authors want to express the concentration of 25-hydroxyvitamin D in blood plasma using the units of ng/ml then different values need to be quoted. Adequate concentrations of 25-hydroxyvitamin D in blood serum is usually specified as greater than 50 nmol/L. This is equivalent to greater than 20 ng/ml. Therefore, the units quoted in the text are quite wrong as ng/ml.
3. Line 33 and in many, many other places throughout the text: The terms 25-hydroxyvitamin D and 25-hydroxyvitamin D3 are used in a very confusing way. In many places throughout the text the term “25(OH)D and D3” is used. This is quite mystifying because vitamin D3 was not measured. It appears that this abbreviation refers to measurements of total 25(OH)D – a mixture of 25(OH)D2 and 25(OH)D3, compared with values for 25(OH)D3 concentration in plasma. This completely wrong term needs to be corrected throughout the manuscript and needs to be expressed as total 25(OH)D and 25(OH)D3 concentrations.
4. Lines 36 and 37: The term: “differences were detected in vitamin D by sex and BMI..” is confusing. What is apparently meant is covered by the term “vitamin D status” When the authors are writing about deficiency or adequate status for vitamin D the word “status” should be used, because the term “vitamin D” alone refers to the parent molecules of cholecalciferol or ergocalciferol.
5. Lines 50-51: “Vitamin D receptors reside in many cells…” There is no mention in the text that these receptors are for the endocrine product of vitamin D – 1,25-dihydroxyvitamin D.
6. Line 52: “Vitamin D metabolic enzymes…..” This term is meaningless in the context it is used. Are these enzymes that catabolise vitamin D? Are they enzymes that have particular roles in the endocrine production of 1,25-dihydroxyvitamin D? This term needs to be explained, or more information needs to be provided, to indicate that vitamin D and 25-hydroxyvitamin D are precursors to the hormone 1,25-dihydroxyvitamin D.
7. Line 137: “d6-25-Hydroxyvitamin D3” It would be helpful to explain to readers that this is a deuterium-labelled standard of 25-hydroxyvitamin D3, as it is not easily understood what “d6” refers to.
8. Lines 331-332: “plasma levels of vitamin D have been shown to be very stable…” This statement is possibly true, but the authors are not referring to the concentration of vitamin D in blood plasma. The statement refers to the concentration of 25-hydroxyvitamin D in blood plasma. This should be made clear by revising this sentence and many others where this mistake occurs.
9. Line 347: “ Vitamin D is a steroid hormone…” This is quite wrong. Vitamin D is a precursor molecule to a steroid hormone: 1,25-dihydroxyvitamin D. Vitamin D itself is not a steroid hormone.
10. Line 374: “vitamin D levels” here, as in many, many other places in the text, the substance being referred to is not vitamin D but is a metabolite of vitamin D, 25-hydroxyvitamin D.
Author Response
Detailed responses to reviewer comments
Reviewer 1
- Line 24: The term “plasma vitamin D levels were quantified”. This is quite wrong as what was measured was 25-hydroxyvitamin D in blood plasma. It is confusing to specify “vitamin D levels” because there is vitamin D in blood as well as 25-hydroxyvitamin D. Therefore, for every measurement provided in this document the term “25-hydroxyvitamin D” should be used and not “vitamin D” because vitamin D concentration was not measured.
REPLY: Thank you for your comment. For consistency, the Vitamin D has been replaced throughout when our reported data is specified with 25-hydroxyvitamin D. With respect to citations/references to the scientific literature, there are times in which Vitamin D is used.
- Lines 25-26: The concentration of 25-hydroxyvitamin D in blood is grouped in the so-called ranges of “<30 ng/ml” “50 ng/ml” and “51-71 ng/ml”. The units here are incorrect. These values are not ng/ml but instead are “nmoles/litre” or “nM” The values expressed in ng/ml to indicate deficiency or sufficiency of vitamin D status are not those quoted. If the authors want to express the concentration of 25-hydroxyvitamin D in blood plasma using the units of ng/ml then different values need to be quoted. Adequate concentrations of 25-hydroxyvitamin D in blood serum is usually specified as greater than 50 nmol/L. This is equivalent to greater than 20 ng/ml. Therefore, the units quoted in the text are quite wrong as ng/ml.
REPLY: For accuracy, we did measure 25-hydroxyvitamin D2 and 25-hydroxyvitamin D3, and have combined the results. The units that we measured with confirmed standards are intended for quantification as ng/ml. The concentration ranges we used for this analysis are based on upper respiratory tract infections (URI) literature. Quraishi et. al reports definition of vitamin D insufficiency as 25(OH)D concentrations less than 30 ng/ml and deficiency below 20 ng/ml. Sabetta et. al. reports maintenance of 25-hydroxyvitamin D serum concentration of 38 ng/ml or higher should significantly reduce the incidence of acute viral respiratory tract infections. We have updated the methods (line 210) and added multiple citations for these reported ranges. The D2 and D3 status are provided in supplementary data.
[20] S. A. Quraishi and C. A. Camargo, “Vitamin D in acute stress and critical illness,” Curr. Opin. Clin. Nutr. Metab. Care, vol. 15, no. 6, pp. 625–634, Nov. 2012, doi: 10.1097/MCO.0b013e328358fc2b.
[21] D. R. Murdoch et al., “Effect of vitamin D3 supplementation on upper respiratory tract infections in healthy adults: the VIDARIS randomized controlled trial,” JAMA, vol. 308, no. 13, pp. 1333–1339, Oct. 2012, doi: 10.1001/jama.2012.12505.
[22] K. Y. Z. Forrest and W. L. Stuhldreher, “Prevalence and correlates of vitamin D deficiency in US adults,” Nutr. Res. N. Y. N, vol. 31, no. 1, pp. 48–54, Jan. 2011, doi: 10.1016/j.nutres.2010.12.001.
[23] J. R. Sabetta, P. DePetrillo, R. J. Cipriani, J. Smardin, L. A. Burns, and M. L. Landry, “Serum 25-hydroxyvitamin d and the incidence of acute viral respiratory tract infections in healthy adults,” PloS One, vol. 5, no. 6, p. e11088, Jun. 2010, doi: 10.1371/journal.pone.0011088.
- Line 33 and in many, many other places throughout the text: The terms 25-hydroxyvitamin D and 25-hydroxyvitamin D3 are used in a very confusing way. In many places throughout the text the term “25(OH)D and D3” is used. This is quite mystifying because vitamin D3 was not measured. It appears that this abbreviation refers to measurements of total 25(OH)D – a mixture of 25(OH)D2 and 25(OH)D3, compared with values for 25(OH)D3 concentration in plasma. This completely wrong term needs to be corrected throughout the manuscript and needs to be expressed as total 25(OH)D and 25(OH)D3 concentrations.
REPLY: Thank you for bringing this to our attention, and the 25(OH)D has been updated to read “total 25(OH)D” throughout the manuscript. We measured both 25-hydroxyvitamin D2 and D3 in our assay. However, in this cohort only 5 participants had 25(OH)D2 at detectable levels above the LOQ (limit of quantification). Notably, this is likely due to the source of vitamin D that the participant obtained from plants (e.g. different forms in plants vs animal sources).
- Lines 36 and 37: The term: “differences were detected in vitamin D by sex and BMI.” is confusing. What is apparently meant is covered by the term “vitamin D status” When the authors are writing about deficiency or adequate status for vitamin D the word “status” should be used, because the term “vitamin D” alone refers to the parent molecules of cholecalciferol or ergocalciferol.
REPLY: Thank you for your comment. Line 39 and throughout the manuscript we have now updated this to read as “status”
- Lines 50-51: “Vitamin D receptors reside in many cells…” There is no mention in the text that these receptors are for the endocrine product of vitamin D – 1,25-dihydroxyvitamin D.
REPLY: This section now reads, “Vitamin D metabolite receptors reside in many cells, and as a whole, the 1, 25-hydroxyvitamin D network impacts a suite of host processes and cellular functions.”
Line 52: “Vitamin D metabolic enzymes…..” This term is meaningless in the context it is used. Are these enzymes that catabolise vitamin D? Are they enzymes that have particular roles in the endocrine production of 1,25-dihydroxyvitamin D? This term needs to be explained, or more information needs to be provided, to indicate that vitamin D and 25-hydroxyvitamin D are precursors to the hormone 1,25-dihydroxyvitamin D.
REPLY: The first paragraph of the introduction has now been revised and this reference to the metabolic enzymes is removed accordingly to reduce confusion. We also determined that an expanded explanation is beyond the scope of findings reported herein. The text now simply introduces the concept for existing relationships with chronic disease risk. “Vitamin D metabolism influences immune cells in a manner that decreases risk for developing chronic diseases such as diabetes and cardiovascular disease.”
- Line 137: “d6-25-Hydroxyvitamin D3” It would be helpful to explain to readers that this is a deuterium-labelled standard of 25-hydroxyvitamin D3, as it is not easily understood what “d6” refers to.
REPLY: For clarification the tex now reads “25-Hydroxyvitamin D2, 25-Hydroxyvitamin D3 and d6-25-Hydroxyvitamin D3 (deuterium labeled internal standard), were obtained from Cerilliant (Round Rock, TX)”
- Lines 331-332: “plasma levels of vitamin D have been shown to be very stable…” This statement is possibly true, but the authors are not referring to the concentration of vitamin D in blood plasma. The statement refers to the concentration of 25-hydroxyvitamin D in blood plasma. This should be made clear by revising this sentence and many others where this mistake occurs.
REPLY: Thank you for this detail. The text has now been revised. “The variability in timing between first and second blood draws may account for this difference because it spans both acute and convalescent phases of infection. Interestingly, the uptake and release of 25(OH)D in skeletal muscle may also contribute to the stability in circulation when measured over several months [11].
Line 347: “ Vitamin D is a steroid hormone…” This is quite wrong. Vitamin D is a precursor molecule to a steroid hormone: 1,25-dihydroxyvitamin D. Vitamin D itself is not a steroid hormone.
REPLY: Thank you. Line 422 has been updated with more clarification and now reads, “Total 25(OH)D is a secosteroid with hormone properties that exhibits activity in all cells through VDR-dependent mechanisms and has an array of impact on genes involved in cell proliferation, differentiation, bactericidal protein production, innate and adaptive immunity.”
- Line 374: “vitamin D levels” here, as in many, many other places in the text, the substance being referred to is not vitamin D but is a metabolite of vitamin D, 25-hydroxyvitamin D.
REPLY: “Vitamin D level” is now updated to read “total 25-hydroxyvitamin D”
Reviewer 2
- Page 8: Figure 3B is not comprehensive. What is the difference between left and right panel of Figure 3b. I guess that the right panel showed the result from patients whose 25(OH) D increased and left showed the data 25(OH)D decreased. However, it was not clear. Moreover, I could not understand how authors picked up the data from 84 patients. authors must add the explanation of them in the result part and legend of Figure 3B.
REPLY: Thank you for bringing this to our attention. Line 299 explains the n=84 patients. Line 302-304 clarified results between the two panels in figure 3B.
- Page10 line 321: DNAm →DNA?
REPLY: Thank you for your comment, line 373 has been updated to “DNA”
Reviewer 2 Report
The manuscript is interesting. However, there are a few issues which author should make it clear.
1. Page 8: Figure 3B is not comprehensive. What is the difference between left and right panel of Figure 3b. I guess that the right panel showed the result from patients whose 25(OH) D increased and left showed the data 25(OH)D decreased. However, it was not clear. Moreover, I could not understand how authors picked up the data from 84 patients. authors must add the explanation of them in the result part and legend of Figure 3B.
2. Page10 line 321: DNAm →DNA?
Author Response
Detailed responses to reviewer comments
Reviewer 1
- Line 24: The term “plasma vitamin D levels were quantified”. This is quite wrong as what was measured was 25-hydroxyvitamin D in blood plasma. It is confusing to specify “vitamin D levels” because there is vitamin D in blood as well as 25-hydroxyvitamin D. Therefore, for every measurement provided in this document the term “25-hydroxyvitamin D” should be used and not “vitamin D” because vitamin D concentration was not measured.
Thank you for your comment. For consistency, the Vitamin D has been replaced throughout when our reported data is specified with 25-hydroxyvitamin D. With respect to citations/references to the scientific literature, there are times in which Vitamin D is used.
- Lines 25-26: The concentration of 25-hydroxyvitamin D in blood is grouped in the so-called ranges of “<30 ng/ml” “50 ng/ml” and “51-71 ng/ml”. The units here are incorrect. These values are not ng/ml but instead are “nmoles/litre” or “nM” The values expressed in ng/ml to indicate deficiency or sufficiency of vitamin D status are not those quoted. If the authors want to express the concentration of 25-hydroxyvitamin D in blood plasma using the units of ng/ml then different values need to be quoted. Adequate concentrations of 25-hydroxyvitamin D in blood serum is usually specified as greater than 50 nmol/L. This is equivalent to greater than 20 ng/ml. Therefore, the units quoted in the text are quite wrong as ng/ml.
For accuracy, we did measure 25-hydroxyvitamin D2 and 25-hydroxyvitamin D3, and have combined the results. The units that we measured with confirmed standards are intended for quantification as ng/ml. The concentration ranges we used for this analysis are based on upper respiratory tract infections (URI) literature. Quraishi et. al reports definition of vitamin D insufficiency as 25(OH)D concentrations less than 30 ng/ml and deficiency below 20 ng/ml. Sabetta et. al. reports maintenance of 25-hydroxyvitamin D serum concentration of 38 ng/ml or higher should significantly reduce the incidence of acute viral respiratory tract infections. We have updated the methods (line 210) and added multiple citations for these reported ranges. The D2 and D3 status are provided in supplementary data.
[20] S. A. Quraishi and C. A. Camargo, “Vitamin D in acute stress and critical illness,” Curr. Opin. Clin. Nutr. Metab. Care, vol. 15, no. 6, pp. 625–634, Nov. 2012, doi: 10.1097/MCO.0b013e328358fc2b.
[21] D. R. Murdoch et al., “Effect of vitamin D3 supplementation on upper respiratory tract infections in healthy adults: the VIDARIS randomized controlled trial,” JAMA, vol. 308, no. 13, pp. 1333–1339, Oct. 2012, doi: 10.1001/jama.2012.12505.
[22] K. Y. Z. Forrest and W. L. Stuhldreher, “Prevalence and correlates of vitamin D deficiency in US adults,” Nutr. Res. N. Y. N, vol. 31, no. 1, pp. 48–54, Jan. 2011, doi: 10.1016/j.nutres.2010.12.001.
[23] J. R. Sabetta, P. DePetrillo, R. J. Cipriani, J. Smardin, L. A. Burns, and M. L. Landry, “Serum 25-hydroxyvitamin d and the incidence of acute viral respiratory tract infections in healthy adults,” PloS One, vol. 5, no. 6, p. e11088, Jun. 2010, doi: 10.1371/journal.pone.0011088.
- Line 33 and in many, many other places throughout the text: The terms 25-hydroxyvitamin D and 25-hydroxyvitamin D3 are used in a very confusing way. In many places throughout the text the term “25(OH)D and D3” is used. This is quite mystifying because vitamin D3 was not measured. It appears that this abbreviation refers to measurements of total 25(OH)D – a mixture of 25(OH)D2 and 25(OH)D3, compared with values for 25(OH)D3 concentration in plasma. This completely wrong term needs to be corrected throughout the manuscript and needs to be expressed as total 25(OH)D and 25(OH)D3 concentrations.
Thank you for bringing this to our attention, and the 25(OH)D has been updated to read “total 25(OH)D” throughout the manuscript. We measured both 25-hydroxyvitamin D2 and D3 in our assay. However, in this cohort only 5 participants had 25(OH)D2 at detectable levels above the LOQ (limit of quantification). Notably, this is likely due to the source of vitamin D that the participant obtained from plants (e.g. different forms in plants vs animal sources).
- Lines 36 and 37: The term: “differences were detected in vitamin D by sex and BMI.” is confusing. What is apparently meant is covered by the term “vitamin D status” When the authors are writing about deficiency or adequate status for vitamin D the word “status” should be used, because the term “vitamin D” alone refers to the parent molecules of cholecalciferol or ergocalciferol.
Thank you for your comment. Line 39 and throughout the manuscript we have now updated this to read as “status”
- Lines 50-51: “Vitamin D receptors reside in many cells…” There is no mention in the text that these receptors are for the endocrine product of vitamin D – 1,25-dihydroxyvitamin D.
This section now reads, “Vitamin D metabolite receptors reside in many cells, and as a whole, the 1, 25-hydroxyvitamin D network impacts a suite of host processes and cellular functions.”
Line 52: “Vitamin D metabolic enzymes…..” This term is meaningless in the context it is used. Are these enzymes that catabolise vitamin D? Are they enzymes that have particular roles in the endocrine production of 1,25-dihydroxyvitamin D? This term needs to be explained, or more information needs to be provided, to indicate that vitamin D and 25-hydroxyvitamin D are precursors to the hormone 1,25-dihydroxyvitamin D.
The first paragraph of the introduction has now been revised and this reference to the metabolic enzymes is removed accordingly to reduce confusion. We also determined that an expanded explanation is beyond the scope of findings reported herein. The text now simply introduces the concept for existing relationships with chronic disease risk. “Vitamin D metabolism influences immune cells in a manner that decreases risk for developing chronic diseases such as diabetes and cardiovascular disease.”
- Line 137: “d6-25-Hydroxyvitamin D3” It would be helpful to explain to readers that this is a deuterium-labelled standard of 25-hydroxyvitamin D3, as it is not easily understood what “d6” refers to.
For clarification the tex now reads “25-Hydroxyvitamin D2, 25-Hydroxyvitamin D3 and d6-25-Hydroxyvitamin D3 (deuterium labeled internal standard), were obtained from Cerilliant (Round Rock, TX)”
- Lines 331-332: “plasma levels of vitamin D have been shown to be very stable…” This statement is possibly true, but the authors are not referring to the concentration of vitamin D in blood plasma. The statement refers to the concentration of 25-hydroxyvitamin D in blood plasma. This should be made clear by revising this sentence and many others where this mistake occurs.
Thank you for this detail. The text has now been revised. “The variability in timing between first and second blood draws may account for this difference because it spans both acute and convalescent phases of infection. Interestingly, the uptake and release of 25(OH)D in skeletal muscle may also contribute to the stability in circulation when measured over several months [11].
Line 347: “ Vitamin D is a steroid hormone…” This is quite wrong. Vitamin D is a precursor molecule to a steroid hormone: 1,25-dihydroxyvitamin D. Vitamin D itself is not a steroid hormone.
Thank you. Line 422 has been updated with more clarification and now reads, “Total 25(OH)D is a secosteroid with hormone properties that exhibits activity in all cells through VDR-dependent mechanisms and has an array of impact on genes involved in cell proliferation, differentiation, bactericidal protein production, innate and adaptive immunity.”
- Line 374: “vitamin D levels” here, as in many, many other places in the text, the substance being referred to is not vitamin D but is a metabolite of vitamin D, 25-hydroxyvitamin D.
“Vitamin D level” is now updated to read “total 25-hydroxyvitamin D”
Reviewer 2
- Page 8: Figure 3B is not comprehensive. What is the difference between left and right panel of Figure 3b. I guess that the right panel showed the result from patients whose 25(OH) D increased and left showed the data 25(OH)D decreased. However, it was not clear. Moreover, I could not understand how authors picked up the data from 84 patients. authors must add the explanation of them in the result part and legend of Figure 3B.
Thank you for bringing this to our attention. Line 299 explains the n=84 patients. Line 302-304 clarified results between the two panels in figure 3B.
- Page10 line 321: DNAm →DNA?
Thank you for your comment, line 373 has been updated to “DNA”
Round 2
Reviewer 1 Report
The units for defining vitamin D status used in this report are the 25-hydroxyvitamin D serum concentrations in ng/ml. This is completely appropriate as either ng/ml or nmoles/L (nM) can be used. The FDA definition of vitamin D status is as follows:
30 nM or less is deficient = 12 ng/ml
30-50 nM is insufficient = 12 – 20 ng/ml
50-75 nM is optimal = 20 - 30 ng/ml
At lines 209-210 the following definitions of vitamin D status are given for 25-hydroxyvitamin D concentration in serum or plasma::
30 ng/ml or less is deficient
30-50 ng/ml is insufficient
51-71 ng/ml is optimal
However, if these values of ng/ml are expressed as nM units the following would apply:
30 ng/ml or less = 50 nM or less is deficient
30-50 ng/ml = 75 – 125 nM is insufficient
51-71 ng/ml = 127.5 – 177.5 nM is optimal
The vitamin D status as defined in this manuscript is completely at odds with the vitamin D status defined by FDA and many other public health authorities throughout the world.
Author Response
Thank you for reiterating the need to check our results and interpretations with respect to the following units used in our analysis, and the accepted federal guidelines for Vitamin D. The manuscript results, figures and text have now been updated throughout with this guidance in our revision.
30 nM or less is deficient = 12 ng/ml
30-50 nM is insufficient = 12 – 20 ng/ml
50-75 nM is optimal = 20 - 30 ng/ml
Reviewer 2 Report
I think new version of the manuscript is acceptable for the journal.
Author Response
Thanks for the review. Please note that based on the prior reviewer comment, substantial edits have now been made to the results and the interpretations are now using the following guidelines.
30 nM or less is deficient = 12 ng/ml
30-50 nM is insufficient = 12 – 20 ng/ml
50-75 nM is optimal = 20 - 30 ng/ml